# Unveiling connectivity patterns of railway timetables through complex network theory and Infomap clustering

**Fabio Lamanna**[1]*, **Michele Prisma**[2], **Giorgio Medeossi**[2]

**1** Freelance Civil Engineer, Treviso, Italy, **2** Trenolab, Gorizia, Italy

* fabio@fabiolamanna.it

## Abstract

This study introduces a novel framework for analysing railway timetable connectivity through complex network theory and the Infomap clustering algorithm. By transforming timetable data into network representations, we systematically assess the connectivity and efficiency of the Norwegian railway system for the 2024 and projected 2033 scenarios. We define and implement the Timetable Connectivity Index ($T_c$), a comprehensive metric integrating service frequency, travel times, and the hierarchical network structure. The analysis is conducted across three distinct network spaces: Stops, Stations, and Changes, with both unweighted and weighted networks. Our results reveal key insights into how infrastructural developments, service frequencies, and travel time adjustments influence network connectivity. Comparative analysis of the two scenarios highlights improvements in physical infrastructure and travel time efficiency by 2033, while also identifying areas where service frequency distribution and transfer effectiveness may require further optimization. The findings provide valuable insights for railway planners and operators, aiming to improve the efficiency and reliability of train networks.

## Introduction

Railway timetable research focuses on optimizing connectivity and performance indices to enhance the efficiency and reliability of train networks. Researchers analyse how timetable design impacts connectivity, which refers to how well different stations are linked and the ease of transferring between trains. They also assess performance indices such as punctuality, frequency, and travel time. By examining these factors, researchers aim to develop schedules that minimize delays, improve synchronization between trains, and ensure seamless passenger transfers [1]. Advanced techniques, including mathematical modelling and simulation, are used to evaluate different timetable scenarios, ultimately guiding decisions that improve overall service quality and operational efficiency in railway systems [2]. The assessment of railway timetable performance and structure has primarily involved simulation approaches, utilizing various indicators and quality levels [3]. More recent studies have focused on expected passenger travel times using macroscopic models [4]. Other research explores the interplay between timetable design and robustness [5], whereas some methodologies assess robustness by identifying critical points within the network [6] or by passengers' perspective and

**Data availability statement:** All code and example data necessary to reproduce our analyses have been deposited in a public Github

Repository (https://github.com/fblamanna/TimetableNetworkTool). The repository contains the full Python scripts for generating directed, weighted networks in the Spaces of Stations, Stops and Changes (including both DSN and DTN weights), the sample input timetable CSV, and the output Pajek ".net" files as input to Infomap analysis. All materials are released under the GNU General Public License and may be freely downloaded, inspected, and reused.

**Funding:** The author(s) received no specific funding for this work.

**Competing interests:** The authors have declared that no competing interests exist.

disutility [7]. A recent study emphasizes low-cost, high-impact adjustments suitable for real-world implementation of robustness, increasing dwell times and headways at key points in the network [8], by means of RCP (Robustness at Critical Point) indicators. Reliability of timetables has been also explored by timetable heterogeneity indices [9]. Alternative indicators of performance in transit systems include also entropy and betweenness centrality to assess the efficiency of public transport networks [10]. Taking a different perspective, research on complex networks has dramatically increased in recent years. Complex networks are defined as systems composed of many elements (nodes and edges) interacting with each other. Edges are often associated with weights representing the flow of information through the network, influenced by both topology and the probabilistic characteristics of the weights [11]. Theories and algorithms for complex networks span biological, social, technological, and transportation domains [12]. The analysis of complex networks in transportation systems draws on foundational discoveries in complex network theory, such as the emergence of scaling properties [13] and the characterization of highly clustered systems with small characteristic path lengths, known as small-world networks [14]. These theories highlight how information can be transmitted quickly through networks in relatively few steps. One of the early studies applying these principles to railway networks was conducted on the Boston subway system [15]. Authors introduced the concept of network efficiency, defined as the measure of how effectively a network exchanges information. Their findings indicated that the Boston subway system exhibits small-world characteristics with high communication efficiency. Similarly, the Indian Railway Network (IRN) was analysed [16], introducing the notion of a "link" as a connection between nodes based on train services that stop at various stations. Traditionally, networks have been represented by binary edges (i.e., present or absent). The introduction of weighted complex networks analysis in transportation systems [17] allows for a more nuanced analysis by incorporating the strength of connections, such as the number of services operating between stations within a given time frame [18] to take into account traffic dynamics. This approach was advanced [19] by analysing railway networks through multiple layers, including changes, stops, and stations, extracting the topology of the Swiss railway network from timetable data, where edge weights represent traffic flows. The latter provided a topological characterization using statistical indices, such as clustering coefficients and average path lengths, as well as distributions of node degrees (the number of services starting or arriving at each station) and edge weights (services on links). The multilayer framework over network analysis grew in the last years, including analysis on transportation networks [20] which naturally fit into this schema, especially when dealing with multimodal systems. Regarding railway systems, a recent study [21] applies complex network theory to assess connectivity improvements through timetable adjustments, highlighting measures of service frequency and travel time adjustments taking care of network characterization. Further studies have increasingly explored the application of clustering and modularity-based methods in transportation networks to better organize transit systems [22] or demonstrating how community detection techniques can uncover latent urban structures and network hierarchies based on transit flows [23]. Transportation networks, in general, are characterized by both infrastructural layers and flows (passengers, number of trains running, freights, etc.) that characterized the full information of the system with dynamics and pure topological structure [24]. In this paper, having the interweaving behaviour of flows and infrastructure in memory, we explore differences in connectivity and performance across various timetables using Infomap [25], a clustering algorithm that identifies community structures in networks by minimizing the description length of a random walker's path. Infomap leverages information theory to efficiently partition the network into densely connected clusters, making it particularly effective for analysing large and complex networks where flows have a strong importance. Unlike

prior work that largely relies on simulation or macroscopic modeling, our approach transforms timetables into network spaces—Stops, Stations, and Changes—and applies Infomap to extract hierarchical community structures based on both topology and flow. This enables the definition of the Timetable Connectivity Index, a metric that captures the modular organization and functional efficiency of timetable scenarios. Our contribution lies in combining flow-based clustering with a structural performance metric, offering planners an interpretable and scalable tool for comparing and optimizing timetable designs.

## Materials and methods

The framework we developed starts from the analysis of timetable data in *.csv* format, following this general schema (for more details see S1 Table in S1 File of Supporting information):

- Train number
- Station
- Arrival time
- Departure time
- Stop type

The "Stop type" field indicates whether a station is a scheduled stop or not (e.g., stop/pass/service), while the meanings of the other fields are easily discernible. The timetable data cover passenger operations within a time frame (peak hour, working day, etc.). For each train, the list enables us to create a sequence of stations with the respective arrival and departure times. Before conducting the analysis on the real dataset, we first applied a synthetic approach to assess and validate our methodology. This preliminary step was crucial for ensuring the robustness of our method and for clarifying its application for the reader. By using a controlled simple synthetic dataset, we were able to systematically test the various components of our approach, identify potential issues, and refine our techniques before applying them to the more complex real-world data. This process not only strengthens the validity of our findings but also enhances the reader's understanding of how the methodology works in a simplified context.

### From timetables spaces to networks

The first step in our methodological analysis of timetables involves transforming each train route, including departure and arrival times at stations, into a network. To accomplish this, we first define the "Spaces" for our analysis. Based on the methodology described earlier in the paper [19], we introduce three different network systems or Spaces:

1. **Space of Stops**: here, two stations are connected if they are consecutive stops on at least one train route. From a network perspective, there may be some links (or shortcuts) that bypass stations not part of the train route (Stop type = pass);
2. **Space of Stations**: in this Space, two stations are connected only if they are linked by a physical track. This Space represents the topological network of stations and their connections;
3. **Space of Changes**: in this Space, two stations are connected if there is at least one service that stops at both stations. All stations within a single train route are fully interconnected, forming a clique (a subset of nodes where every pair of nodes is directly connected by an edge). Different cliques are linked by stations that serve as nodes for possible service interchanges. In other words, all stations served by a single train service are

interconnected, indicating that a passenger can travel between these stations without needing to change trains.

Fig 1 illustrates the concepts discussed using a simplified railway network with three different passenger services: one express and two regional services. This figure models the following Spaces:

1. **Space of Stops**: this Space shows the direct connections between trains;
2. **Space of Stations**: this Space reflects the network's infrastructure topology;
3. **Space of Changes**: this Space depicts two primary network structures interconnected by a single node (station B), which serves as an interchange point for services among lines.

The networks obtained from these analyses include multiple links between nodes due to the various services operating on each branch (i.e., the timetable Space generates Multigraphs). The next step is to simplify these networks into a single Weighted Graph, where each link between nodes is associated with information derived from the timetable. To achieve this, we consider two types of network and weights:

1. **DSN (Directed Service Network)** with weights as the total number of services running within each Space across the entire timetable period, between each couple of nodes; here we expect a network of local, operational modules (strong service-based clusters);
2. **DTN (Directed Travel Time Network)** with weights evaluated as the average travel time derived from the timetable, between each couple of nodes; here we expect a broader, accessibility-based system (that may skip intermediate stops). In the DTN, the weight assigned to each directed edge is computed as the inverse of the average travel

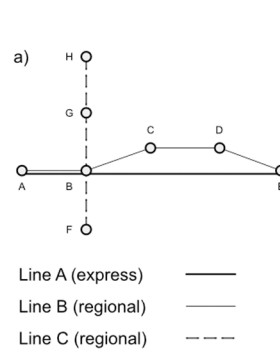

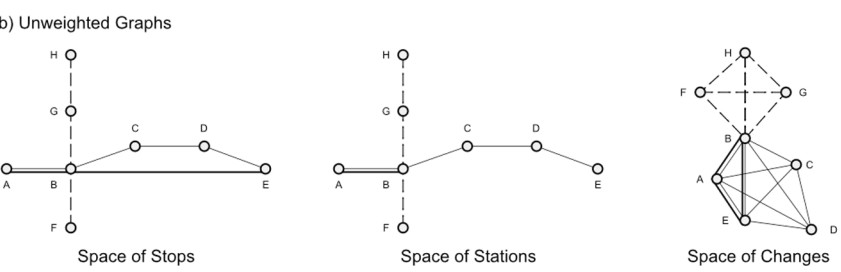

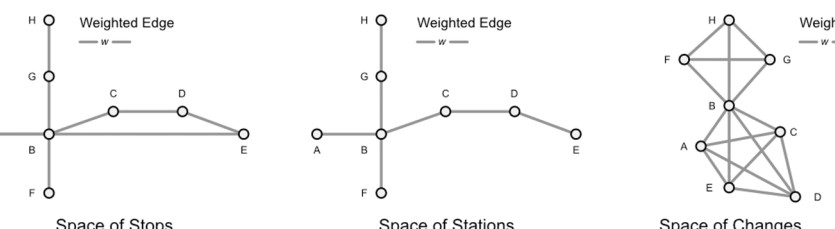

**Fig 1. From timetable spaces to networks.** We developed a synthetic model to test the full framework. The model begins with a network of three train services: Line A (express), Line B (regional), and Line C (regional) (a), connecting eight nodes (stations) labeled from A to E. We first construct unweighted networks (b) as multi-edge graphs, where each edge represents a single train service. Finally, we incorporate weighted functions in the DSN (number of services) and DTN (travel time) representations between each pair of nodes (c). In this final scenario, multiple edges are flattened into a single weighted graph, where the edge weight $w$ stores information about the number of services or travel times between nodes.

time $t$ between two nodes across all relevant services. This inverse is applied linearly (i.e., $w = \frac{1}{t}$), so that faster connections yield higher weights, emphasizing their greater contribution to network performance.

This approach helps flatten the network, making it easier to analyse and interpret the data. In the end, we obtained three different networks for each weight type, resulting in a total of six weighted networks representing the timetable information. Each network is directed, meaning that each link between stations has a specific direction, allowing for another link in the opposite direction to account for services traveling in the reverse direction. In network theory terms, this is equivalent to an undirected graph with two distinct weights, one for each direction.

## Clustering framework and Infomap

Once the network representations of the railway timetable are prepared, we apply clustering algorithms to uncover structural patterns based on both topology and edge weights. To gain meaningful insights from these networks, it is crucial to utilize quantitative measures that account for the complexity of connections and service frequencies. Traditional methods for identifying community structures in directed and weighted networks often simplify the problem by disregarding the directions and weights of links. Such approaches, while useful in some contexts, overlook significant information about the network's structure. In railway networks, where connections and service frequencies play a vital role, ignoring these aspects can lead to incomplete or misleading conclusions about the network's organization and performance. By incorporating edge weights into the analysis, we can better capture the flow patterns and uncover meaningful structures within the timetable network. To address this limitation, we seek a methodology that integrates both topology and edge weights into the analysis. Flow-based approaches, such as those derived from the map equation, are well-suited for this purpose. In our study, we employ the Infomap algorithm, which excels in identifying community structures in complex networks by minimizing the description length of a random walker's path. Infomap is a sophisticated clustering algorithm used to uncover the community structure in complex networks. It leverages the concept of information theory to efficiently partition a network into clusters or communities. In the Infomap algorithm, "flow" refers to the movement of information or resources through a network, which is modelled as a random walker's trajectory. The probability of the walker moving from one node to another is determined by both the topology and the weights of the edges, where higher weights indicate a higher likelihood of movement along that path. For more detailed explanations about the algorithm, please refer to the above cited paper. This method effectively captures both the topology and the weighted connections, offering insights into the intrinsic structure of the network that might be missed by simpler methods, capturing the persistence time of the so-called random walker within a certain structure (cluster). We selected Infomap over other popular clustering algorithms such as Louvain or Leiden because it is particularly well-suited for flow-based, directed, and weighted networks—characteristics that align closely with our representation of timetable connectivity. Unlike modularity-based methods, Infomap leverages information theory to capture dynamic flow structures, making it especially effective in identifying communities in transportation networks where edge direction and flows (e.g., service frequency or travel time) are essential.

## Levels, flows and timetable characteristics

The main goal of the framework is to well characterise the networks to get effective insights about clustering formation within timetable systems. First, we define the Levels of analysis: they refer to the hierarchical structure that the Infomap algorithm can uncover within a network. The algorithm doesn't just identify flat, single-level communities; it can also reveal multiple levels of nested communities, providing a more detailed and hierarchical view of the network's structure. Here's how these levels work:

- First Level - Primary Communities: at the most basic level, Infomap identifies the primary communities within the network. These are groups of nodes that are more densely connected to each other than to the rest of the network;
- Second Level - Sub-Communities: within each primary community, Infomap can further partition the nodes into sub-communities. These sub-communities represent a finer level of structure, where the nodes are even more tightly connected to each other than they are to other nodes within the same primary community. This level captures the internal organization of larger communities;
- Higher Levels - Deeper Hierarchical Structure: the process can continue to higher levels, identifying nested sub-communities within sub-communities, depending on the complexity of the network. Each subsequent level provides a more granular view of the network's hierarchical structure.

Just as a cartographer adjusts the scale of a map to determine which details are included - omitting minor streets on a regional map that would be highlighted on a city map - the appropriate size or resolution of modules in the timetable network analysis depends on the scope of the stations and connections included. In our approach, this concept is mirrored in the Infomap algorithm. For instance, at a higher level, Infomap might identify broad clusters of stations connected by major routes, akin to a regional railway map highlighting key intercity connections. At a finer resolution, Infomap can reveal detailed sub-clusters within these larger groups, like how a city map might detail every local station and track within an urban rail network. The level of detail in these modules adapts to the universe of nodes in the network, just as a map's detail is tailored to its scale. For this work, we focus on Primary Communities only (i.e. Level 1), going deeper into the hierarchical structure of the system only for data validation purposes. From now on, we will refer to clusters as Modules. Secondly, we define the flow within each Level: the flow is significantly influenced by the weights of the edges, which represent the strength of connections between nodes, in terms of number of services or average travel time; in this last case, as stated earlier in the paper, we built the weight as the inverse of the pure travel time, so to give more importance to edges having fast connections. A Module might represent a regional network within the national system, where trains primarily circulate within a certain area, or areas connected by fast services. This clustering technique helps in understanding how different parts of the network function semi-independently yet are connected to the larger system. Flows are normalized to 1 within each Level of the hierarchical structure obtained by Infomap, giving the fraction of importance of each Module within the system. Fig 2 shows the results obtained by applying the above framework to the synthetic network. We define the number of services operating during the analysis period (e.g., peak hour) and construct the directed weighted network in the Space of Stops accordingly. We then apply Infomap to this network, identifying two primary Modules at the first level, which include stations predominantly served by express services (2 trains/hour) within the timetable system. Each primary Module is further divided into sub-levels, which captures the remaining connections among stations within the Module (e.g. the first main Module comprises six

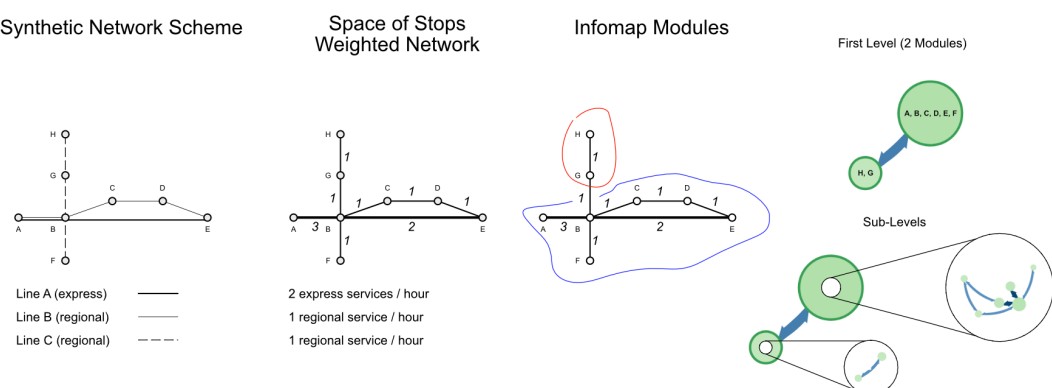

**Fig 2. Infomap modules and levels.** We perform analysis on the synthetic network creating the first Modules by applying the Infomap clustering algorithm to the timetable; results can be derived from multiple Levels, going deeper into the clustering formation process.

nodes, each one within its own flow, as represented in Table 1). Sample results show that Station B is, as expected, the key node in the system; it serves as an exchange node, with both express services and regional trains passing through it. After testing the framework on a simple synthetic model, we need to perform quantitative analysis on a real case scenario; to do so, we introduce a Timetable Connectivity Index function of the information derived from Infomap and Timetable Spaces.

## Timetable connectivity index

The Timetable Connectivity Index ($T_c$) is a comprehensive measure designed to evaluate the overall connectivity of a railway network based on its timetable structure. This index considers several key factors that contribute to the connectedness of the network. First, the total number of modules ($M$) within a Level plays a crucial role; generally, fewer modules indicate a more connected timetable, as it suggests that stations are grouped into larger, more cohesive clusters. Within each module, the flow ($F_m$) represents the frequency of services or fast connected group of stations, highlighting the module's importance to the network's overall

**Table 1. Synthetic network – flows on nodes and modules.**

| Node | Module Level Path | Node-Level Flow | Module Flow |
|------|-------------------|-----------------|-------------|
| B | 1:1 | 0,364 | 0,864 |
| A | 1:2 | 0,137 | |
| E | 1:3 | 0,137 | |
| D | 1:4 | 0,091 | |
| C | 1:5 | 0,091 | |
| F | 1:6 | 0,045 | |
| G | 2:1 | 0,091 | 0,136 |
| H | 2:2 | 0,045 | |

For each node in the network, Infomap returns the Level Path tree along with the Flow associated to each single node and with the full Module. In practical terms, the Module Flow quantifies the relative importance of each cluster within the network based on the cumulative flow of services it handles. A higher Module Flow indicates that the corresponding cluster of stations plays a central role in service distribution. Similarly, Node-Level Flow values reflect how critical individual stations are within their modules: for example, Station B, with the highest Node-Level Flow, acts as a major interchange or hub, supporting both regional and express services. These values provide a quantitative insight into the hierarchy of service importance, helping to identify core nodes for operational planning and resilience analysis.

connectivity. Finally, the number of nodes per module ($N_m$) reflects how many stations are connected within each main module, further influencing the network's connectedness. The index is calculated using the following formula where $N$ is the total number of nodes in the network.

$$T_c = \frac{1}{N} \sum_{m=1}^{M} N_m F_m,$$ (1)

Additionally, the distribution of flow within nodes in a module can offer insights into the relative importance of individual stations based on the amount of traffic they handle. However, our analysis of real data indicates that the impact of flow distribution within modules on overall connectivity is negligible (see S1 Fig in S1 File of Supporting information). By summing over modules within a single Level, $T_c$ provides an overall measure of how well-connected the timetable is, considering both the internal structure of the network and the flow of trains. In general, higher values of $T_c$ indicates a more connected and efficiently structured timetable, with strong flows within well-defined modules across different levels; lower $T_c$ values might suggest that the network is either poorly connected or lacks well-defined hierarchical structures, indicating potential areas for improvement in the timetable design. This index ranges from 0 to 1 and provides a quantitative measure that integrates the detailed information captured by the Infomap algorithm, offering valuable insights into the connectivity and efficiency of the railway timetable. For the sake of clarity, we perform some basic analyses on the synthetic network shown in Fig 3. We begin with a basic network in the Space of Stops, featuring three different services (two regional and one express, as previously illustrated in Fig 2) and the corresponding edge weights, which represent the number of trains running within a fixed time frame. First, we evaluate the Timetable Connectivity Index in scenario a), considering the full network; two main Modules emerge, primarily due to the shortcut link between stations A and E. In scenario b), the express link between stations B and E is removed. As a result, Infomap splits the system into three modules (one more module compared to the first case). $T_c$, as expected, is significantly lower in this scenario because the removal of a critical link reduces overall connectivity within the timetable system. In scenario c), we introduce an additional direct service between stations A and C, with a single train service during the time frame ($w = 1$). $T_c$ increases slightly, reflecting the improved connections within the main module. Finally, in scenario d), we retain the new link but increase the weight (i.e., the number of trains running) from 1 to 10, and we evaluate the corresponding $T_c$ values. The results show that the module partition remains unchanged, while the global $T_c$ increases to just under 0.5, despite the higher number of services. This outcome is consistent with the observation that, although the new link has a significantly higher number of trains, the overall connectivity improvement within the timetable system is marginal. This is because $T_c$ not only accounts for the quantity of services (edge weights), but also how such services affect the modular structure of the network. Since the additional weight does not lead to a reconfiguration of modules or the emergence of new interconnections beyond the existing cluster, the structural change is limited. The module partition remains unchanged, meaning that the added frequency reinforces an existing connection without significantly enhancing overall network connectivity. This highlights how $T_c$ captures structural—not merely numerical—improvements, emphasizing the importance of strategically placed connections over volume alone. So far, the Timetable Connectivity Index has been evaluated only at the First Level (main modules) due to the simplicity of the network and flows being considered. In general, it can be applied at each Level, providing more detailed information as it delves deeper into the hierarchical structure of the system. Since $T_c$ is normalized by the total number of nodes in the network, we can use it for comparisons across networks of different sizes.

Timetable Connectivity Index

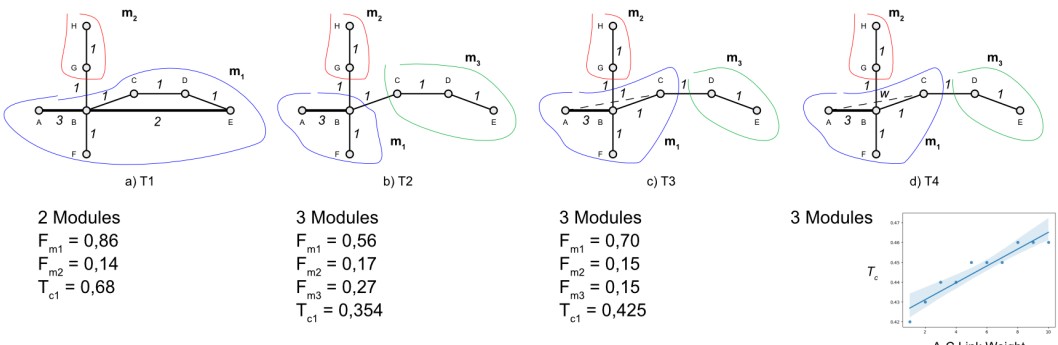

2 Modules
$F_{m1} = 0,86$
$F_{m2} = 0,14$
$T_{c1} = 0,68$

3 Modules
$F_{m1} = 0,56$
$F_{m2} = 0,17$
$F_{m3} = 0,27$
$T_{c1} = 0,354$

3 Modules
$F_{m1} = 0,70$
$F_{m2} = 0,15$
$F_{m3} = 0,15$
$T_{c1} = 0,425$

3 Modules

**Fig 3. Timetable connectivity index** ($T_c$). Analysing a synthetic timetable network with changes allowed us to comprehend the quantitative and qualitative meaning of $T_c$ and its relation with the Module formation process.

However, comparability is most meaningful when the underlying assumptions—such as the selected network space, weighting method (DSN or DTN), and modular level—are consistent across scenarios or systems.

## Results

We apply all the previous framework to Norwegian Timetables, to the aim of comparing two different scenarios: the timetable structure as at the year 2024 (in the following R24) and the planned situation in 2033 (R33) as a "Service Concept". All the analyses have been performed thanks to the Infomap Python API and its web interface [26], the latter mainly to obtain the clustering visualizations. Since the model is probabilistic, we run the model ten times taking the best solution in terms of clustering partition. Data have been generated by Treno software (developed by Trenolab) and derives from the public Norwegian railway data, according to the previous cited format. We also further investigate correlations among number of stations and Modules both in the first and in the second Level of clustering.

### Norwegian timetables spaces

Applying the complex network framework to the Norwegian railway dataset for 2024 enabled us to construct and analyze distinct network representations for different Spaces. Each Space - Stop, Station, and Changes - offers a unique perspective on the railway system. Fig 4 illustrates these networks, with nodes positioned according to geographical coordinates to better visualize the system's spatial layout. The comparison of network structures across different Spaces reveals insightful qualitative variations. In the Space of Stops, the network reflects the actual points where trains halt, providing a view of service coverage. The Space of Stations, incorporating physical infrastructure, offers a more detailed representation, showing how stations are interconnected through the physical railway lines. Here, the network's shape shifts slightly, highlighting the distribution and connectivity of stations. The Space of Changes presents a distinct view, emphasizing the connections and transfers between different services. This Space reveals the emergence of visual clusters, particularly around major railway hubs where direct services converge. These clusters indicate areas of high connectivity and suggest the locations of significant urban and suburban interchange points. This spatial organization underscores the importance of these hubs in facilitating efficient travel within the

Norwegian Railway Timetable (2024)

**Fig 4. Network representations of the Norwegian railway timetable (2024) across three spaces: stops, stations, and changes.** We apply the framework to the Norwegian Timetable as in 2024; the northern isolated branch linking Narvik with the Swedish border has been excluded from the analysis (since it is physically disconnected from the whole Norwegian Network and it does not add meaningful information about the full connectivity). Node and edge sizes are uniform, as weighted information is not yet encoded at this stage of the analysis. The figure is intended to highlight the spatial structural differences between Spaces. Darker regions in the layout indicate higher node or edge density, typically corresponding to urban hubs or areas with multiple closely spaced stations (e.g., Oslo, Bergen, Trondheim, Stavanger). In the Space of Stops, certain edges act as Shortcuts Service links, connecting non-consecutive stations served by express trains. In the Space of Stations, each edge corresponds to a physical Infrastructure link. The Space of Changes reflects full interconnections among stations served by the same train (Direct Service links), forming tightly clustered structures around main transfer points.

network. Timetable Connectivity Indices (evaluated on the first Modular Level only) are presented in Figs 5 and 6, grouped by Weighted Network (Services – DSN, Travel Time - DTN) and Spaces (Stations, Stops, Changes) for each scenario (R24, R33). In the analysis of the railway timetables for the scenarios of 2024 (R24) and 2033 (R33), several key insights can be drawn by examining the Timetable Connectivity Index ($T_c$) across different network spaces, both in unweighted and weighted cases. The indices reveal how infrastructural changes, service frequency, and travel times impact overall network connectivity. Full results for the first Level of analysis are available on S2 Table in S1 File of Supporting information.

**Correlations among number of stations and module flow:** The analysis of the relationship between the number of nodes within each module and the corresponding module flow revealed a strong positive and statistically significant correlation across all network spaces and clustering levels (see S2 Fig in S1 File of Supporting information). The consistent positive correlation observed between module size and module flow across different network representations (DSN and DTN) and spatial abstractions (Stations, Stops, Changes) reveals fundamental organizational patterns in railway timetable structures. In service-based networks (DSN), the correlation is stronger, particularly at the first modular level, reflecting how operational decisions naturally concentrate services in station-dense areas, reinforcing strong regional hubs. In contrast, travel time-based networks (DTN) show slightly weaker but still significant correlations, highlighting a more dispersed pattern of accessibility, where efficient travel is not solely dependent on the number of served stations. From a management perspective, these findings imply that service expansions—such as adding new train routes or increasing frequencies—will naturally amplify existing hubs unless strategic actions are taken to enhance connectivity in peripheral regions (e.g. with new fast services taking advantage of new lines in the forecasted Scenario R33).

**Space of Stations – Unweighted case:** the unweighted analysis of the Space of Stations focuses solely on the infrastructural aspects of the network, disregarding service frequency

## Directed Service Network (DSN)

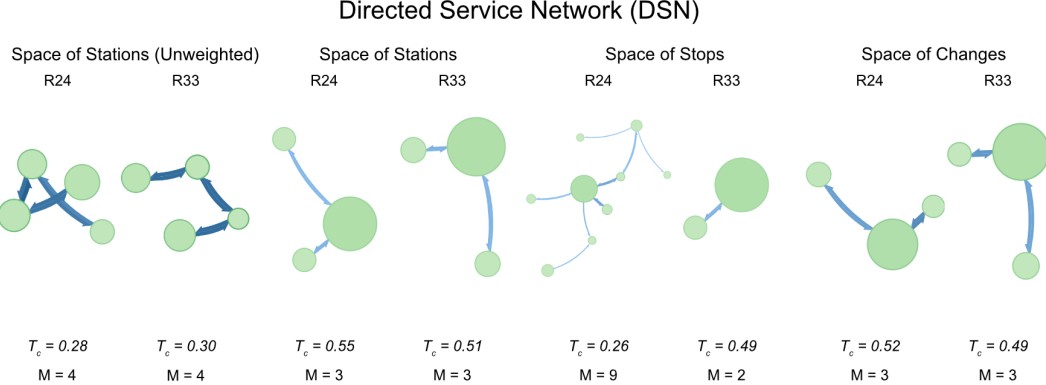

**Fig 5. Directed Service Network (DSN) – Comparison of modular structures and Timetable Connectivity Index ($T_c$) for R24 and R33 across the three Spaces (Stations, Stops, Changes).** The diagrams illustrate how service frequency affects the modular structure of the network. In R33, several modules become denser in the Space of Stops, reflecting more frequent train stops and improved accessibility, especially in urban and regional corridors. In contrast, a slight decrease in $T_c$ in the Space of Stations suggests a more uneven distribution of services across the expanded infrastructure. These patterns underscore the need for balanced service planning to fully exploit infrastructural investments.

## Directed Travel Time Network (DTN)

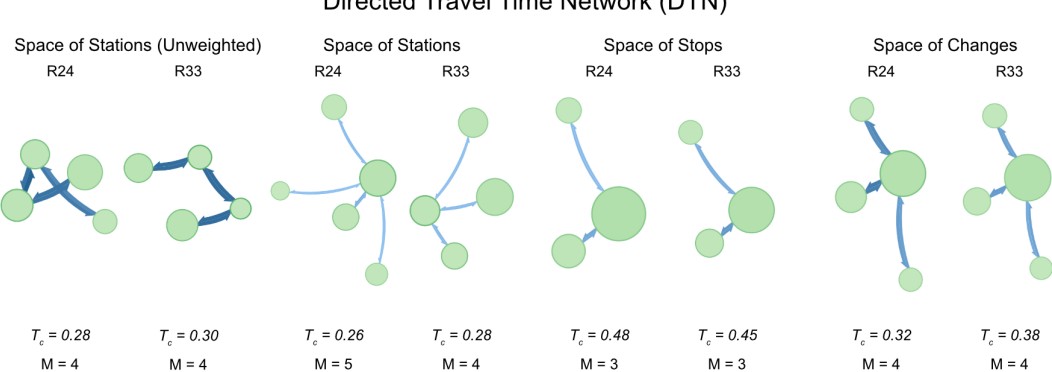

**Fig 6. Directed Travel Time Network (DTN) – Comparison of modular structures and Timetable Connectivity Index ($T_c$) for R24 and R33.** The increase in $T_c$ in the Space of Stations and Changes in R33 reflects improved travel time efficiency due to infrastructure upgrades and high-speed services. Conversely, the slight drop in $T_c$ in the Space of Stops suggests that despite more services, the efficiency between stops has not improved proportionally—likely due to longer dwell times or added complexity in local services. These results highlight the importance of aligning service patterns with infrastructure design to achieve both accessibility and efficiency.

and travel times. $T_c$ increases from 0.28 in R24 to 0.30 in R33, suggesting an improvement in the underlying infrastructure over the years. This improvement indicates that, by 2033, the railway network's physical connections (i.e., stations and tracks) are slightly better integrated. This increase in $T_c$ (from 0.28 to 0.30) is consistent with the real-world developments, such as the doubling of tracks in the Oslo area and other key regions. For example, the new 200 km/h line from Oslo to Bergen (OSL-HFS-BRG) and the extension of double tracks in various corridors, such as from Drammen (DRM) to Tønsberg (TBG), contribute to better connectivity at the infrastructural level. These changes enhance the physical integration of the network, which is captured by the increase in $T_c$. The width of arrows among nodes is a proxy of how much flow is exchange through Modules.

**Space of Stations – Directed Service Network (DSN):** when weights are introduced based on the number of trains running (DSN), $T_c$ for the Space of Stations slightly decreases from 0.55 in R24 to 0.51 in R33. This decline suggests that, despite infrastructural improvements, the overall service frequency across the network has not been optimized or has possibly become more uneven. The drop in $T_c$ might indicate that some stations receive fewer services or that the distribution of train services has shifted, possibly prioritizing different routes. The slight decrease in $T_c$ (from 0.55 to 0.51) might initially seem counterintuitive, given that R33 includes many more trains in general, including the increase in services on key long-distance corridors like Oslo-Stavanger (OSL-STV), Oslo-Bergen (OSL-BRG), and Oslo-Trondheim (OSL-TND). However, this decrease could be explained by the redistribution of services, where the focus on enhancing regional services and introducing new high-speed lines might lead to a more complex service pattern, which could slightly reduce the overall service frequency when viewed across the entire network. It is useful to remember here that in the Space of Stations there are far more nodes (stations) than in other Spaces, since we are considering the full infrastructure, including 'pass' and service-only stations.

**Space of Stops – Directed Service Network (DSN):** the Space of Stops, which reflects the points where trains actually stop, shows a significant increase in $T_c$ from 0.26 in R24 to 0.49 in R33. This substantial rise suggests a marked improvement in the connectivity of services where they matter most to passengers at the stops. It implies that by 2033, the timetable has been adjusted to ensure more frequent stops or better service distribution, enhancing the network's accessibility and convenience for passengers. The significant increase in $T_c$ for the Space of Stops (from 0.26 to 0.49) correlates well with the real-world enhancements in the Oslo area and other regional corridors. The doubling of services and the introduction of new direct links, especially in areas like Ski (SKI), Høvik (HLD), and Stjørdal (STJ), contribute to much better stop-level connectivity. The extension of regional services, such as those in the Stavanger (STV) and Bergen (BRG) areas, and the intensified services between Stjørdal (STJ) and Trondheim (TND), support this observed improvement in stop-level connectivity.

**Space of Changes – Directed Service Network (DSN):** in the Space of Changes, where the focus is on transfer points between services, $T_c$ decreases slightly from 0.52 in R24 to 0.49 in R33. This marginal decrease might indicate a slight reduction in the ease of transfers, possibly due to changes in service patterns or scheduling that make connections between different services slightly less efficient. This is an area that may require further attention to maintain or improve passenger transfer experiences. The slight decrease in $T_c$ (from 0.52 in R24 to 0.49 in R33) may reflect the redistribution and increase in service complexity due to the new infrastructure and service patterns. While key nodes like Oslo (OSL), Drammen (DRM), and Bergen (BRG) have seen increased services, the introduction of more direct services and new high-speed lines might have reduced the necessity for transfers, thus slightly impacting the overall $T_c$ in the space of changes in the R33 Scenario.

**Space of Stations – Directed Travel Time Network (DTN):** when considering travel time (DTN), $T_c$ for the Space of Stations increases from 0.26 in R24 to 0.28 in R33. This improvement suggests that, by 2033, the average travel time between stations has decreased, indicating faster or more direct services. The enhanced travel time efficiency reflects a better alignment between infrastructure and service delivery, contributing to an overall improvement in network performance. The increase in $T_c$ (from 0.26 in R24 to 0.28 in R33) in the Space of Stations (DTN) aligns well with the introduction of faster travel options, such as the new high-speed line from Oslo to Bergen. The reduced travel times due to these infrastructural upgrades and the focus on high-speed regional services (e.g., SHI-OSL-HFS) have effectively enhanced the network's travel time efficiency, as captured by the increase in $T_c$.

**Space of Stops – Directed Travel Time Network (DTN):** in the Space of Stops, $T_c$ decreases from 0.48 in R24 to 0.45 in R33. This slight decline could indicate that while more stops are being served (as seen in the DSN analysis), the travel time efficiency between these stops has not improved at the same rate. This could be due to increased dwell times at stops (i.e. time spent by a train in a station without moving) or slower services on certain routes, which might offset the benefits of increased service frequency. The slight decrease in $T_c$ between scenarios (from 0.48 to 0.45) could reflect the increased complexity and service patterns, where, despite more frequent stops, the overall travel time efficiency might not have improved proportionally. This might be due to the added services along extended routes, which, while increasing connectivity, do not necessarily reduce travel times significantly across the network.

**Space of Changes – Directed Travel Time Network (DTN):** finally, $T_c$ for the Space of Changes (DTN) increases from 0.32 in R24 to 0.38 in R33, suggesting that the efficiency of travel times at transfer points has improved. This improvement indicates that by 2033, the timetables have been optimized to enhance overall connectivity in terms of total travel time. The increase in $T_c$ in R33 (from 0.32 in R24 to 0.38 in R33) reflects the improved efficiency at key transfer points. This can be attributed to reduced travel times on key corridors (e.g., Oslo to Sweden), which improve the overall connectivity for transfers within the network, as passengers benefit from faster and more efficient connections between major nodes.

**Overall conclusions:** the analysis highlights both improvements and areas needing attention in the railway network from 2024 to 2033. While there are gains in infrastructure (unweighted $T_c$) and travel time efficiency at stations and transfer points (DTN), the slight decline in service frequency-related $T_c$ in certain spaces (DSN) suggests the need for careful consideration in service planning to ensure that enhancements in infrastructure and travel time are complemented by optimal service distribution across the network. The enhancements in infrastructure, service patterns, and travel times across various regions and corridors, particularly around Oslo, Stavanger, and Bergen, are well reflected in the Timetable Connectivity Index values, providing a comprehensive understanding of how these changes impact the overall connectivity and performance of the railway network. Analysis of $T_c$ across different time windows (full-day, peak, and off-peak) reveals that timetable connectivity may vary significantly throughout the day. In general, as expected, $T_c$ is higher during peak hours, reflecting more concentrated services, which typically form well-defined clusters—both in terms of service frequency and travel time. These findings highlight that $T_c$ is not only structurally but also temporally sensitive—capable of detecting how connectivity strength shifts across service windows. Detailed results are available on S3 Fig in S1 File of Supporting information.

## Scope and limitations

The scope of this study focuses on analyzing railway timetable connectivity through the lens of complex network theory, using the Timetable Connectivity Index ($T_c$) and the Infomap clustering algorithm. The proposed framework quantifies the structural connectivity of timetables, emphasizing the availability of connections, the clustering of stations, and the hierarchical organization of services. Specifically, this approach provides insights into timetable design by identifying key connectivity patterns and evaluating differences between current and future scenarios. However, this study has certain limitations. The analysis is primarily based on static timetable data and does not explicitly account for reliability and robustness, such as the ability of the timetable to absorb delays and disruptions. $T_c$ captures the

structural efficiency of connections but it does not directly measure the operational performance of the network under real-world conditions, such as variability in service punctuality or transfer synchronization. Future work could integrate additional metrics to address these aspects, providing a more comprehensive view of timetable resilience. The availability of trains on various routes, influencing the frequency of connections and waiting times, is indirectly represented in the analysis through weighted network models. However, the framework does not explicitly address waiting time optimization since, so far, transportation demand is not included as input data in the analysis, and we are not able to perform Origin/Destination assessment within the timetable structure. In summary, this study presents a novel methodology for uncovering connectivity patterns in railway timetables, offering a valuable tool for evaluating and improving timetable structures. While the framework provides a robust foundation, future refinements could address reliability, robustness, and operational factors, thereby expanding its applicability to real-world network optimization challenges.

## Future work and framework extensions

While the present study is based on static timetable data, future work could integrate real-time or probabilistic travel times to assess robustness and reliability under operational variability. This could be achieved by constructing new network layers, such as a Robustness Travel Time Network (RTN), where edge weights reflect observed or simulated delays. Robustness could then be measured in terms of expected increases in passenger disutility [7], incorporating variables such as waiting times, number of transfers, and missed connections. From an operator's perspective, the analyses could be further refined using established reliability indicators such as the Scheduled Speed Heterogeneity Ratio (SSHR) and Scheduled Arrival Headway Regularity (SAHR) [9]. Embedding such metrics within the clustering framework would allow for the identification of structurally vulnerable regions in the timetable, where reliability is compromised due to flow or network design limitations. Once tested, these enhancements might extend the current approach toward a more comprehensive evaluation of timetable resilience. Finally, it's important to point out that the current framework is designed for application to single-mode networks, where travel time and service frequency metrics are internally consistent. Extension to multimodal networks is conceptually feasible—particularly given that Infomap supports multilayer analysis—but requires careful treatment of intermodal transfers, harmonization of service frequency scales, and consideration of additional transfer penalties. These methodological challenges represent valuable directions for future research to enhance the framework's applicability to more complex, integrated mobility systems.

## Conclusion

This study presented a comprehensive framework for analyzing railway timetable connectivity using complex network theory and the Infomap clustering algorithm. By transforming railway timetables into network representations, we examined the connectivity and efficiency of the Norwegian railway system for the years 2024 and 2033. Our approach, which integrates topology and edge weights, allowed us to uncover underlying structural patterns within the network, providing a nuanced understanding of how infrastructural developments, service frequencies, and travel time adjustments influence overall network connectivity over time. The strong positive correlations observed between module size and module flow across all network spaces emphasize the inherent structural coherence of railway timetable networks and offer valuable guidance for future service planning and accessibility improvements. The application of the Timetable Connectivity Index ($T_c$) revealed key insights into the evolving

connectivity of the Norwegian railway network. Our findings indicate that while infrastructural improvements led to enhanced physical connections between stations, the optimization of service frequency and travel times remains crucial for maximizing network efficiency. The analysis showed that although some increases in connectivity were observed (e.g., an increase in $T_c$ from 0.26 to 0.49 in the Space of Stops from R24 to R33), the redistribution of services and the introduction of new routes occasionally resulted in more complex service patterns that did not always correlate with improved overall connectivity. Beyond theoretical contributions, the framework developed in this study holds concrete value for railway operators and infrastructure planners, emphasizing the critical role of both infrastructural and operational factors in shaping railway timetable connectivity and performance. The Timetable Connectivity Index ($T_c$) offers a practical metric to evaluate and compare the effectiveness of different timetable configurations, supporting evidence-based decisions for service planning. By revealing how connectivity evolves under infrastructure or service changes, $T_c$ can guide targeted improvements such as increasing service frequencies on weakly connected modules or enhancing interchange stations. The Infomap-derived modular structures further enable the identification of functionally cohesive sub-networks, helping planners to detect emerging hubs or underserved regions. These insights are directly applicable to strategic planning, timetable redesign, and the evaluation of long-term transport policies. Finally, the Timetable Connectivity Index could be easily integrated into existing scheduling and diagnostic platforms; by embedding $T_c$ into the workflows, operators could supplement traditional KPIs (e.g., punctuality, dwell time) with structural insights on how changes affect modular connectivity, supporting scenario comparison within the platform.

## Acknowledgments

Authors thank all the people at Trenolab for their support and contribution to this paper. F.L. would also like to thank attendees at the 11th International Conference on Railway Operations Modelling and Analysis (RailDresden 2025), where the research has been presented for the first time to an audience, for their useful comments and discussions.

## Supporting information

**S1 File. Pdf file containing the SI.** This file includes 3 figures (S1 Fig: Distribution of Interquantile Range within Modules; S2 Fig: Scatterplots showing the relationship between module size ($N_m$) and module flow for different modular structures of the Norwegian railway timetable network; S3 Fig: Sensitivity Analysis of $T_c$ across full day, peak and off-peak time windows) and 2 tables (S1 Table: Timetable data: general input schema; S2 Table: Norwegian Timetable; first Level of clustering results).
(PDF)

## Author contributions

**Conceptualization:** Fabio Lamanna.

**Data curation:** Michele Prisma.

**Formal analysis:** Fabio Lamanna.

**Investigation:** Fabio Lamanna, Michele Prisma, Giorgio Medeossi.

**Methodology:** Fabio Lamanna, Michele Prisma, Giorgio Medeossi.

**Software:** Fabio Lamanna.

**Supervision:** Michele Prisma, Giorgio Medeossi.

**Validation:** Michele Prisma.

**Writing – original draft:** Fabio Lamanna, Michele Prisma, Giorgio Medeossi.

**Writing – review & editing:** Fabio Lamanna, Michele Prisma.

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
