## [Decision Letter · Decision Letter 0]

31 May 2025

PONE-D-25-21673Unveiling connectivity patterns of railway timetables through complex network theory and Infomap clusteringPLOS ONE

Dear Dr. Lamanna,

Thank you for submitting your manuscript to PLOS ONE. After careful consideration, we feel that it has merit but does not fully meet PLOS ONE’s publication criteria as it currently stands. Therefore, we invite you to submit a revised version of the manuscript that addresses the points raised during the review process.

We look forward to receiving your revised manuscript.

Kind regards,

Yong-Hong Kuo

Academic Editor

PLOS ONE

Journal Requirements:

2. We note you have included a table to which you do not refer in the text of your manuscript. Please ensure that you refer to Table 1 in your text; if accepted, production will need this reference to link the reader to the Table.

Additional Editor Comments :

The referees have provided a series of suggestions to improve the work. Two critical issues are the practicality of the work and also the presentation. The authors are encouraged to address the concerns seriously in the revision.

Reviewers' comments:

Reviewer's Responses to Questions

**Comments to the Author**

1. Is the manuscript technically sound, and do the data support the conclusions?

Reviewer #1: No

Reviewer #2: Yes

Reviewer #3: Yes

Reviewer #4: Yes

2. Has the statistical analysis been performed appropriately and rigorously? 

Reviewer #1: No

Reviewer #2: Yes

Reviewer #3: No

Reviewer #4: N/A

3. Have the authors made all data underlying the findings in their manuscript fully available?

Reviewer #1: No

Reviewer #2: Yes

Reviewer #3: No

Reviewer #4: Yes

4. Is the manuscript presented in an intelligible fashion and written in standard English?

Reviewer #1: No

Reviewer #2: Yes

Reviewer #3: Yes

Reviewer #4: Yes

5. Review Comments to the Author

Reviewer #1: Quality of paper is very poor

Quality of picture, managing paper is very bad and resubmit

I suggest to retype paper with consulting professional writer

Please use

https://www.researchgate.net/publication/369618584_Standard_format_for_writing_paper_in_industrial_engineering_optimization_web_of_science_journal

Page size: A4 and 21*29.7

Header from top: 1.27 cm

Footer from bottom: 1.27 cm

Margin: normal 2.54 from left, right, top, bottom

Font: Times New Roman Size:12

Reviewer #2: Firstly, this manuscript investigates connectivity patterns in railway timetables using complex network theory and Infomap clustering, specifically applied to Norwegian railway scenarios for 2024 and projected conditions for 2033. The central objective, clearly articulated in the abstract (page 1), aims at understanding how service frequencies, travel times, and hierarchical structures affect overall network connectivity. The abstract is well-written, concise, and informative, although I suggest explicitly mentioning the key comparative insights between the two analyzed scenarios for greater impact.

The introduction (pages 1-2, lines 1-60) effectively sets the research context, outlining key concepts such as connectivity, service optimization, and robustness in railway timetabling. However, the section could significantly benefit from a more explicit statement regarding the novelty and contributions of this specific study, differentiating it clearly from previous research mentioned on lines 12-16 (papers [1]-[4]). Cite this “By examining these factors, researchers aim to develop schedules that minimize delays, improve synchronization between trains, and ensure seamless passenger transfers” as https://doi.org/10.1109/ICIRT.2016.7588740 . Cite this as “Complex networks are defined as systems composed of many elements (nodes and edges) interacting with each other.” as https://doi.org/10.1016/j.ress.2021.108130 . While relevant studies are referenced, a succinct paragraph clearly articulating this study’s unique methodological contribution—particularly the application of Infomap clustering—is strongly advised for clarity.

The methodological approach (pages 2-6, lines 61-260) is comprehensive and robust. The authors explain their three distinct “Spaces” (Stops, Stations, Changes) clearly. The synthetic model explanation on page 3 (lines 82-108) is well-structured and helpful in clarifying the methodological process. Figure 1 (page 3) is particularly effective in illustrating the conceptual differences among the three “Spaces”. However, the caption of Figure 1 lacks details explaining colors or specific node labels within the figure; this should be improved for immediate clarity.

In the section describing the transformation of timetables into weighted graphs (lines 109-127), the distinction between Directed Service Network (DSN) and Directed Travel Time Network (DTN) is clear. However, I recommend explicitly defining how exactly the travel time weights were computed—whether the inverse travel time metric used is linear or otherwise weighted should be made explicit. This point currently lacks sufficient detail.

Regarding the Infomap clustering description (lines 128-156), the manuscript adequately explains why Infomap is appropriate. Still, I recommend briefly mentioning why Infomap was chosen over other popular clustering algorithms (e.g., Louvain or Leiden), ensuring transparency of methodological choice. Table 1 (page 5) is clearly presented but could be better integrated with the surrounding explanatory text, directly interpreting the significance of the module flow and node-level flow values provided in practical terms.

The “Timetable Connectivity Index” (Tc), introduced on pages 6-7 (lines 213-260), is a valuable innovation. The illustrative example in Figure 3 (page 7) effectively demonstrates the practical implications of this metric. However, scenario “d” in Figure 3, involving the increased weight from 1 to 10, requires a clearer textual explanation (lines 250-256) to illustrate why the increase in connectivity is described as marginal despite the substantial numeric change in the number of services.

The Results section (pages 7-10, lines 261-410) provides robust empirical insights. Figure 4 (page 8) clearly demonstrates the spatial differences between the three Spaces. Yet, it lacks specific descriptions or a more detailed legend clarifying the meaning of node size or arrow thickness explicitly, and this should be improved to enhance readability.

Figures 5 and 6 (pages 9-10) comparing the Directed Service Network (DSN) and Directed Travel Time Network (DTN) scenarios for R24 and R33 provide crucial insights. However, captions are insufficiently detailed, particularly lacking explicit interpretative statements of what changes between scenarios imply for railway connectivity and operational planning. The textual discussion (lines 290-410) clearly identifies critical changes between 2024 and 2033 but occasionally lacks precise numerical evidence when stating increases or decreases in Tc values. I strongly recommend that these numerical changes are explicitly stated in parentheses next to qualitative claims (e.g., “increase from 0.28 to 0.30”) to improve clarity and reinforce analytical rigor.

Regarding “Scope and Limitations” (lines 411-436), the authors appropriately acknowledge limitations related to the static nature of the data and the exclusion of reliability metrics. Nevertheless, explicitly suggesting potential methodologies to incorporate reliability measures or robustness indicators in future studies would significantly strengthen the discussion.

The Conclusion section (lines 437-462) succinctly summarizes the key findings but lacks explicit mention of concrete implications for practice or specific recommendations for railway operators. I advise adding a paragraph explicitly linking study results to practical timetable improvements, enhancing the applied relevance of the paper.

Finally, I examined references (pages 12-14, lines 1-14). Overall, references are appropriately selected and appear relevant. However, consistency in citation formatting needs substantial improvement to comply with journal guidelines. Several references lack in literature review (I suggest the two: please look above, check all lines to see any place requiring for being cited) and some references lack DOI numbers (for example, references [1]) and uniform formatting (reference [10] missing page ranges), requiring thorough rechecking and revision. In conclusion, I believe that this manuscript presents a valuable methodological innovation in railway timetable analysis with clear practical potential. The strengths include a solid conceptual framework, effective use of Infomap clustering, and insightful empirical analysis comparing different future scenarios. Critical improvements required for this manuscript involve clearer explicit articulation of the novelty, improved methodological detail (especially weights and Infomap choice), more precise and informative figure/table annotations, explicit numerical references to enhance analytical clarity, and stringent adherence to citation standards. These substantial improvements will significantly enhance the manuscript’s clarity, methodological rigor, and applicability, making it a robust contribution to transportation systems research.

Reviewer #3: The article is well drafted, and to the point, however I have certain suggestions for the authors:

1. The abstract and the conclusion can be improved by adding some numbers from the results.

2. The Scope and Limitations section can be narrowed down.

3. The authors can add some data in form of a table to make it easier for the readers to understand what they are working on.

Reviewer #4: Thank you for the submission. The study introduces a useful framework combining complex network theory with Infomap clustering to analyze railway timetables. The Timetable Connectivity Index (Tc) is an interesting contribution, and the real-world application to Norwegian scenarios adds practical relevance.

1. The paper would benefit from more clarity on practical applicability. How would this metric inform real-world decisions by operators or planners? Could it integrate into existing scheduling or diagnostic tools?

2. Please also expand on the transferability of the framework. Would this approach remain effective in larger, denser, or multimodal networks? A discussion on its limitations or adaptability beyond the Norwegian context would add value.

3. The literature review should be broadened. More recent studies on clustering in transit systems, robustness in timetables, and alternative performance metrics would better situate this work within current research.

A few brief questions for the authors:

Why was Infomap chosen over other clustering methods ?

How sensitive is Tc to different time windows (e.g., peak vs. off-peak)?

Is Tc comparable across networks of different sizes, or does it require normalization?

I recommend revision to address these points. With clearer context, broader references, and practical discussion, the manuscript will make a strong contribution.

6. PLOS authors have the option to publish the peer review history of their article (what does this mean?). If published, this will include your full peer review and any attached files.

Reviewer #1: No

Reviewer #2: No

Reviewer #3: No

Reviewer #4: No

---

## [Author Response · Author response to Decision Letter 1]

16 Jun 2025

We uploaded the "Response to Reviewers" file that address all the comments raised during the revision process.

---

## [Decision Letter · Decision Letter 1]

9 Jul 2025

Unveiling connectivity patterns of railway timetables through complex network theory and Infomap clustering

PONE-D-25-21673R1

Dear Dr. Lamanna,

We’re pleased to inform you that your manuscript has been judged scientifically suitable for publication and will be formally accepted for publication once it meets all outstanding technical requirements.

Kind regards,

Yong-Hong Kuo

Academic Editor

PLOS ONE

Additional Editor Comments (optional):

The authors have successfully addressed the referees' concerns. I recommend Accept.

Reviewers' comments:

Reviewer's Responses to Questions

**Comments to the Author**

1. If the authors have adequately addressed your comments raised in a previous round of review and you feel that this manuscript is now acceptable for publication, you may indicate that here to bypass the “Comments to the Author” section, enter your conflict of interest statement in the “Confidential to Editor” section, and submit your "Accept" recommendation.

Reviewer #2: All comments have been addressed

Reviewer #4: All comments have been addressed

2. Is the manuscript technically sound, and do the data support the conclusions?

Reviewer #2: Yes

Reviewer #4: Yes

3. Has the statistical analysis been performed appropriately and rigorously? 

Reviewer #2: No

Reviewer #4: N/A

4. Have the authors made all data underlying the findings in their manuscript fully available?

Reviewer #2: No

Reviewer #4: Yes

5. Is the manuscript presented in an intelligible fashion and written in standard English?

Reviewer #2: Yes

Reviewer #4: Yes

6. Review Comments to the Author

Reviewer #2: Thank you for your thoughtful and constructive response to the review. The revisions you’ve implemented have clearly strengthened the manuscript.

Reviewer #4: all the comments has been replied to, good luck with your future research work.

7. PLOS authors have the option to publish the peer review history of their article (what does this mean?). If published, this will include your full peer review and any attached files.

Reviewer #2: No

Reviewer #4: No

---

## [Editor Report · Acceptance letter]

PONE-D-25-21673R1

PLOS ONE

Dear Dr. Lamanna,

I'm pleased to inform you that your manuscript has been deemed suitable for publication in PLOS ONE. Congratulations! Your manuscript is now being handed over to our production team.

Kind regards,

on behalf of

Dr. Yong-Hong Kuo

Academic Editor

PLOS ONE